# Parental Vaccine Preferences for Their Children in China: A Discrete Choice Experiment

**DOI:** 10.3390/vaccines8040687

**Published:** 2020-11-16

**Authors:** Tiantian Gong, Gang Chen, Ping Liu, Xiaozhen Lai, Hongguo Rong, Xiaochen Ma, Zhiyuan Hou, Hai Fang, Shunping Li

**Affiliations:** 1Centre for Health Management and Policy Research, School of Public Health, Cheeloo College of Medicine, Shandong University, Jinan 250012, China; tian_gchn@163.com (T.G.); liuping_sdu@163.com (P.L.); 2NHC Key Lab of Health Economics and Policy Research, Shandong University, Jinan 250012, China; 3Centre for Health Economics, Monash Business School, Monash University, Melbourne 3145, Australia; gang.chen@monash.edu; 4School of Public Health, Peking University, Beijing 100083, China; laixiaozhen@pku.edu.cn; 5China Center for Health Development Studies, Peking University, Beijing 100083, China; hgrong@bjmu.edu.cn (H.R.); xma@hsc.pku.edu.cn (X.M.); 6School of Public Health, Fudan University, Shanghai 200032, China; zyhou@fudan.edu.cn; 7Peking University Health Science Center-Chinese Center for Disease Control and Prevention Joint Center for Vaccine Economics, Beijing 100083, China; 8Key Laboratory of Reproductive Health National Health Commission of the People’s Republic of China, Beijing 100083, China

**Keywords:** discrete choice experiment, vaccine, parental preference

## Abstract

Vaccination is one of the most cost-effective health investments to prevent and control communicable diseases. Improving the vaccination rate of children is important for all nations, and for China in particular since the advent of the two-child policy. This study aims to elicit the stated preference of parents for vaccination following recent vaccine-related incidents in China. Potential preference heterogeneity was also explored among respondents. **Methods:** A discrete choice experiment was developed to elicit parental preferences regarding the key features of vaccines in 2019. The study recruited a national sample of parents from 10 provinces who had at least one child aged between 6 months and 5 years old. A conditional logit model and a mixed logit model were used to estimate parental preference. **Results:** A total of 598 parents completed the questionnaire; among them, 428 respondents who passed the rational tests were analyzed. All attributes except for the severity of diseases prevented by vaccines were statistically significant. The risk of severe side effects and protection rates were the two most important factors explaining parents’ decisions about vaccination. The results of the mixed logit model with interactions indicate that fathers or rural parents were more likely to vaccinate their children, and children whose health was not good were also more likely to be vaccinated. In addition, parents who were not more than 30 years old had a stronger preference for efficiency, and well-educated parents preferred imported vaccines with the lowest risk of severe side effects. **Conclusion:** When deciding about vaccinations for their children, parents in China are mostly driven by vaccination safety and vaccine effectiveness and were not affected by the severity of diseases. These findings will be useful for increasing the acceptability of vaccination in China.

## 1. Introduction

Vaccination is one of the most cost-effective ways to avoid disease. Currently, it can prevent 2–3 million deaths per year, and a further 1.5 million could be protected if the global coverage of vaccinations was improved [1]. Routine vaccination for children is one of the most successful strategies to ease the burden of infectious diseases [2]. Improving the vaccination rate of children is important for all nations, and for China in particular since the advent of the two-child policy.

There is still a large gap between actual vaccination coverage and the goal [3]. In China, several vaccines are mandatory for children and covered by China’s National Immunization Program (NIP); e.g., the diphtheria–tetanus–acellular-pertussis (DTaP) vaccine, measles–mumps–rubella (MMR) vaccine and hepatitis B vaccine. There are also recommended but not mandatary vaccines, such as rotavirus vaccine, seasonal influenza vaccine and pneumococcal conjugate vaccine. Mandatory vaccines are free to the public and government-funded, whilst recommended vaccines are normally self-paid by parents. The current uptake of recommended vaccines has been estimated to be low in China, at about 6% and 0.7% for seasonal influenza vaccine and 13-valent pneumococcal conjugate vaccine, respectively [4,5]. A report authored by the World Health Organization (WHO) has shown that vaccine hesitancy, as one of the 10 threats to global health in 2019, has impeded the progress made in tackling vaccine-preventable diseases. Vaccine hesitancy, which is defined as the “delay in acceptance or refusal of vaccination despite the availability of vaccination services” [6], has a direct influence on vaccination rate, and a quarter to a third of US parents were affected by this [7,8].

The reasons for vaccine hesitancy are complex. The literature suggests that the key factors that contribute to vaccine hesitancy include the unnaturalness of vaccination [7], heuristic thinking [9] and a loss of public confidence [10]. In China, several vaccine incidents—e.g., the Changchun Changsheng vaccine incident and Shandong illegal vaccine sales—have occurred in the past few years, which may have resulted in a loss of public confidence in vaccines. The Changchun Changsheng vaccine incident involved (i) manufacturing and selling substandard DTaP vaccines, and (ii) the illegal production of freeze-dried rabies vaccines [11], while in the Shandong illegal vaccine sales incident, questionable vaccines (i.e., produced by licensed manufacturers but not transported or stored properly) were sold to 24 provinces and cities without approval [12]. A study conducted in 2018 found that a majority of the respondents held negative attitudes towards vaccines after the Changchun Changsheng vaccine incident [13]. Another study evaluating the impact of Shandong illegal vaccine sales arrived at a similar conclusion [12].

In this context, it is important to understand parental attitudes and preference for vaccines and to explore key factors associated with parents’ decisions to vaccinate their children. A discrete choice experiment (DCE) technique based on random utility theory has been widely applied to study vaccine preference globally, and substantial heterogeneities exist among the findings [14,15,16,17,18]. In mainland China, very limited DCE studies have been conducted regarding the preference for specific or general vaccines, and they have all been constrained to a single province [19,20]. This is the first DCE study to target a national sample with respondents recruited from 10 provinces in China.

The present study had two objectives: (i) to provide insights into the importance of determinants in parental vaccination choices and (ii) to explore preference heterogeneity among parents with different characteristics.

## 2. Method

### 2.1. Discrete Choice Experiment

The discrete choice experiment has been increasingly used in health economics and health service research as a method to elicit participants’ preferences. DCE can also be used to estimate participants’ willingness to pay as well as to predict participation rates given a set of characteristics of goods or services [21,22]. This approach is derived from random utility theory, where participants would choose the option with the highest utility from the alternatives presented [23]. The DCE design and analysis were conducted following the checklist and reports of the International Society for Pharmacoeconomics and Outcomes Research (ISPOR) Conjoint Analysis Task Forces [24,25,26].

### 2.2. Study Population and Sample Size

To ascertain a national parental preference, a multistage sampling design was used. Firstly, 10 provinces/municipalities were selected based on the Division of Central and Local Financial Governance and Expenditure Responsibilities in the Healthcare Sector released by the State Council in 2018, which divided the 31 provinces/municipalities in mainland China into five layers. According to their geographical location and level of economic development, 10 provinces/municipalities were randomly chosen to represent the eastern region (Shandong and Shanghai), western region (Gansu and Chongqing), southern region (Yunnan and Guangdong), northern region (Beijing and Jilin) and central region (Henan and Jiangxi) (Figure 1). Next, except for three municipalities (Beijing, Shanghai and Chongqing), in each of the other seven provinces, one provincial capital and one non-provincial capital city were chosen to balance the regional disparity. Finally, parents with at least one child aged between 6 months and 5 years old were invited to participate in this survey at community healthcare centers or stations. Only one participant per household could take part in this study.

The guidelines proposed by Johnson and Orme suggested that the sample size can be calculated using the equation N > 500 c/(t × a), where c indicates the number of analysis cells, t refers to the number of choice tasks and a is the number of alternatives [27]. In the main-effects only design, c is equal to the largest number of levels among different attributes in the DCE. In our study, the corresponding values for c, t and a are 4, 10 and 2, respectively; therefore, N can be estimated as (500 × 4)/(10 × 2) = 100. Considering the potential regional heterogeneity, a minimum of 100 respondents would need to be recruited in each region [22,28]. In practice, we intended to survey 60 parents in each province and 120 parents in each region.

### 2.3. Survey Development

Based on previously published literature regarding DCE studies on vaccination [14,16,17,29,30], 11 attributes were initially identified. To assess the appropriateness of attributes and levels to be included and to further reduce the number of attributes in our DCE, four experts with several years of vaccination experience were interviewed face-to-face in Jinan Maternity and Childcare Hospital. Two focus groups (*n* = 12) were also conducted. One focus group included four parents only, and the other contained a vaccinologist, three parents and four health economics/DCE experts. They were asked to review and rank a list of potential attributes. Finally, six attributes were selected for this study (Table 1).

A D-efficient design was developed using Ngene software (www.choice-metrics.com), which yielded 60 choice sets that were further divided into six blocks to reduce respondents’ cognitive burden. To check for internal consistency, one choice set in each block was duplicated and was not excluded in the analysis. Each participant received one block randomly and was asked to answer 11 choice sets.

A pairwise two-stage response DCE design was used to maximize the information gained from the respondents [31]. In the first stage, the participants were forced to choose between two alternative vaccination profiles. Then, they were asked to confirm whether they would vaccinate their preferred option from the first stage for their children. An example of a final choice set was shown in Table 2.

In addition to DCE questions (which were presented in a hardcopy questionnaire), the participants’ and their children’s socio-demographic characteristics were also collected using an iPad. Before completing DCE questions, respondents were asked to rate the importance of six attributes. A pilot was conducted among 15 parents in Beijing and Jinan in July 2019 to examine the acceptability, comprehensibility and validity of the experiment. A few modifications were implemented based on feedback from the pilot.

### 2.4. Data Collection and Analysis

The survey was conducted between August and October 2019. Data were collected by means of one-on-one face-to-face interviews with parents waiting for a routine vaccination for their children or remaining for observation after vaccination. Parents are required to take children to vaccination sites for mandatory vaccines, and high vaccination rates have been achieved for these mandatory vaccines; i.e., rates of above 95% have been achieved for DTaP and hepatitis B vaccines [32]. Thus, the potential sample selection bias for this recruitment strategy was low. Before enrolling in the survey, the conditions were explained in detail by interviewers who received specific training by the research team. All participants signed an electronic informed consent form ahead of enrolment and all responses were anonymous. The study received ethics approval from the Peking University Ethics Committee (IRB00001052-19076).

Responses to the hardcopy DCE questionnaire were double-entered into EpiData 3.1 software and then matched with other socio-demographic characteristics obtained from the iPad for processing and analyzing. Descriptive statistics were reported first. Student’s t-test, the χ^2^ test and Wilcoxon rank-sum test were used to compare means and proportions between subgroups depending on the nature of data.

Regarding DCE data, the personal out-of-pocket cost was coded for linearity, and the effect for the severity of diseases prevented by vaccines was likely to be non-linear. When the latter was coded as three separate parameters, the result was similar and the model performance worsened considerably (see Appendix A). Thus, we decided to treat this attribute as a continuous variable, and the remaining attributes were coded as dummy variables.

The goodness of model fit was guided by the Akaike information criterion (AIC) and Bayesian information criterion (BIC) [22,33]. An initial exploratory analysis was conducted using a conditional logit model (see Appendix A), where the preference among respondents was assumed to be homogenous. Mixed logit or latent class models are commonly used to explore preference heterogeneity [23]. We employed the mixed logit model, where the preference was assumed to follow a normal distribution and the coefficient of attribute level was composed of a mean coefficient as well as a standard deviation [26]. In addition, observed variables such as age, relationship with children, education level and working status were also used to estimate the influence on preferences by including a series of interaction terms with attribute levels. All statistical analyses were conducted using Stata 12.1 software.

## 3. Result

### 3.1. Study Population

In total, 598 parents from 10 provinces participated, and 18 parents were excluded from the analysis due to failing to complete the majority of the questionnaire. Among the remaining 580 parents, the mean age was 31 years (standard deviation: 0.21 years), and the mean age of their children was 2 years old. The majority of respondents were mothers (82%) and more than half resided in an urban area (61%)—which was close to the proportion (60%) of urban population in China [34]—had a Bachelor’s degree or above (56%) and were in employment (67%).

Regarding the internal consistency check within the DCE section, 428 (74%) respondents passed the test. There were no significant differences in socio-demographic characteristics between respondents who failed the test, excluding the gender and health status of children—for more details, see Table 3.

### 3.2. Importance Rating

The results of the importance rating are presented in Figure 2; the respondents were asked to rank attributes from the most to the least important aspects. Overall, parents attached the greatest importance to the severity of diseases prevented by vaccines, followed by the protection rate and risk of severe side effects. The out-of-pocket cost and location of the vaccine manufacturer were less important.

### 3.3. Results of DCE Analysis

Regarding the DCE analysis, those who passed the internal consistency test were included in the main analysis, and the mixed logit estimates are presented in Table 4 [30,35]. The full sample results are comparable with the main analysis and are shown in Appendix A. The coefficient of non-vaccination was included to consider the unconditional choice scenario of allowing for opting out. This was significantly negative, which suggests that, on average, parents preferred to vaccinate their children. The estimated preference for the attributes was consistent with our expectations, except for the severity of diseases prevented by vaccines attribute, which was statistically insignificant.

Parents were more likely to choose the vaccines with a higher protection rate, a longer duration of the illness being prevented by the vaccine and a lower risk of severe side effects. The negative coefficient for the location of the vaccine manufacturer suggested that domestic vaccines were preferred to imported vaccines. The negative coefficient of the out-of-pocket cost attribute indicated that a cheaper vaccine would be preferred. The relative importance of the change in the risk of severe side effects from highest to lowest was 1.667 at most, followed by the highest protection rate. Reducing the risk of severe side effects from high to low could yield 2.6 (1.667/0.642) times as much utility, increasing the duration of the illness being prevented by vaccines from 1 to 10 years. Compared to vaccination safety, the duration was less important.

Some estimated standard deviations were significant, indicating the existence of preference heterogeneity. Social–demographic characteristics were compared with attribute levels to examine preference heterogeneity (Table 5). For the non-vaccination interaction terms, the significantly negative coefficients indicated that the subgroup were less likely to choose non-vaccination (i.e., more likely to vaccinate their children). We found that fathers (β = −1.576) or rural parents (β = −1.283) prefer to vaccinate their children, and children whose health was not good were also more likely to be vaccinated. For the other interaction terms, the significantly positive coefficients suggested that the attributes were more important; well-educated parents preferred imported vaccines (β = 0.468) and the vaccines with the lowest risk of severe side effects (β = 0.445). The highest protection rate was valued higher by parents who were not more than 30 years old, and fathers had a stronger preference for a longer protection duration. Other observed characteristics including the working status of parents, whether the parents had a single child, and the gender of children had no significant influence.

## 4. Discussion

This study reported the results of a DCE study into parental vaccine preferences for their children. Some previous DCE studies in vaccines have been constrained [19,20,36,37] to one particular province or special administrative region in China. To the best of our knowledge, this is the first study to survey parents nationwide to explore their vaccine preferences and examine whether preference heterogeneity existed among participants with various characteristics using discrete choice experiments.

Our study found that a minority (12.1%) of parents chose not to vaccine their children in secondary tasks. The significantly negative coefficient of non-vaccination in the mixed logit model confirmed this finding. The preference for non-vaccination has mixed support in the literature. Although most studies found the same result [38,39,40], a study of parents in the Netherlands found that, on average, parents preferred not to vaccinate their children against human papillomavirus [41].

Among all attributes, the risk of severe side effects and the protection rate of the vaccine had the largest effect on vaccination choice. These findings were in line with other vaccine DCE studies. In a study of HPV vaccines in the US in 2010, greater efficacy was the most desired feature and was strongly valued by mothers [40]. A DCE study conducted in the Philippines found that efficacy was valued most as a factor when deciding to vaccinate with leptospirosis vaccines [42]. In a study of pediatric influenza vaccine, parents placed more importance on the risk of side effects [43]. In addition, other studies found that willingness to vaccinate was closely related to vaccination safety and efficacy [38,44]. In China, the first Vaccine Administration Act voted by the Standing Committee of the National People’s Congress in 2019 stated that a compensation system should be implemented for abnormal responses to vaccination, and the bearer of compensation costs depended on whether the vaccine involved was covered by the government-funded Expanded Program on Immunization [45]. Meanwhile, a quality analysis report in the case of abnormal reactions to vaccines should be submitted to the Medical Products Administrations [45]. However, the database is not publicly available, which might be one reason why safety was the decisive factor. The findings suggest that the safety and efficacy of vaccines would be key characteristics influencing parental vaccination decision-making.

The out-of-pocket cost was found to be less important than other significant attributes. Even though several previously published studies indicated that cost was assigned great importance when deriving preferences [30,46,47], the results are incomparable with our study due to differences in the targeted vaccines and targeted population. Another DCE study conducted in China found that the cost was not associated with a stated preference for a vaccine [20]. In China, common vaccines are affordable, which could be supported by the comparison between the household income and the out-of-pocket cost of vaccines shown by the data in our study; e.g., the highest out-of-pocket cost accounts for about 2% of monthly income. This finding suggests that changing the price may not be an effective or optimal method to improve vaccination coverage.

However, the severity of diseases prevented by vaccines (in terms of mortality) was not a significant contributor to a parental preference for a vaccine, which was inconsistent with the result of the importance rating. Some DCE studies in other countries obtained contrasting results, showing that disease severity played an important role when respondents chose a vaccination profile [15,48,49]. An explanation for the results in our study could be that parents lacked medical knowledge and were less sensitive to the differences in levels of disease severity as a result of the current status of the immunization service in China. Insensitivity to disease severity could also be caused by the larger number of attributes included, meaning that participants ignored this attribute. This finding also suggests that it may be not effective to stress the severity of targeted diseases when health workers recommend vaccines to parents.

Somewhat surprisingly, the results of our study show that, a year after the Changchun Changsheng vaccine incident, a domestic vaccine was preferred to an imported vaccine. A similar finding was obtained from a DCE study conducted in Shanghai a year before the Changchun Changsheng vaccine incident [20], even though both studies varied in terms of study populations and study settings. The reasons that parents preferred domestic vaccines could be that domestic vaccines were thought to be more effective [50] and more accessible. The other potential reason is that the regulatory environment is more stringent. Indeed, a public consultation was facilitated after the incident in 2018 [51], and the first Vaccine Administration Act in 2019 was adopted, which aimed to tighten vaccine regulation [45]. Further studies are warranted to better understand the influence of the manufacturer’s location in vaccine preference.

Concerning preference heterogeneity, we found that vaccination preference differed significantly according to the type of dwelling place, the relationship with children, and the health state of children. In addition, other observed variables (age and education level) had a significant influence on the preference for attribute levels. A study from Poland and Hungary found that working mothers placed less weight on effectiveness and illness severity than non-working mothers [15]. Veldwijk et al. also found that respondents with a lower education level and lower health literacy attached more importance to a vaccine with higher effectiveness [52]. Exploring preference heterogeneity for vaccines would be meaningful and helpful for policy-makers to take pertinent measures in different groups.

The present study had several limitations. First, there may be some omitted factors concerning parental preference for vaccination. However, the process for identifying and selecting attributes has followed the recommended guidelines. Second, recent guidance recommends the use of natural frequencies to present risks. Nonetheless, we opted to use terms such as “low”, “moderate” and “high” to describe this attribute, and participants may have interpreted the levels differently, which might have caused an estimation bias. Finally, the results of the DCE and importance rating are not entirely consistent. This suggests that it would be better to reduce respondents’ difficulty in understanding the meaning of attributes by using figures.

## 5. Conclusions

This study used the well-established DCE technique to investigate vaccine preference among parents in China, and it also revealed preference heterogeneity among respondents. On average, parents were more likely to vaccinate their children. A vaccine with high effectiveness and low risk of severe side effects would be more desirable and decisive, while the severity of targeted diseases had little effect on parents’ decisions. Significant preference heterogeneity was identified among respondents. The findings from this study will be helpful for policymakers to implement more effective policy implementation to improve the vaccine uptake rate in China.

## Figures and Tables

**Figure 1 vaccines-08-00687-f001:**
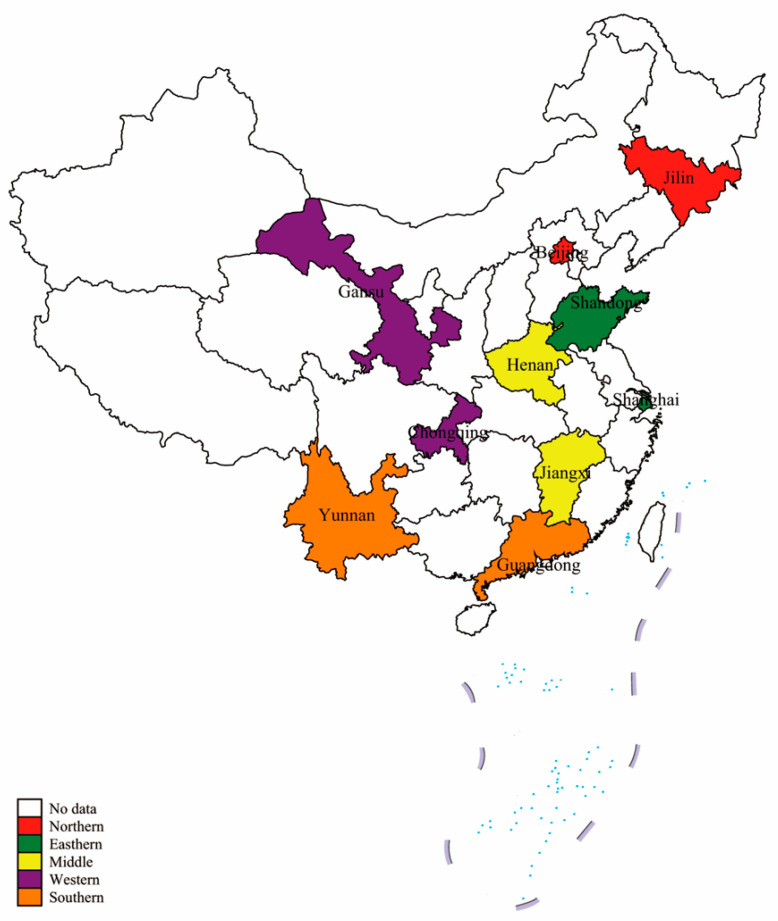
Provinces/municipalities selected in China.

**Figure 2 vaccines-08-00687-f002:**
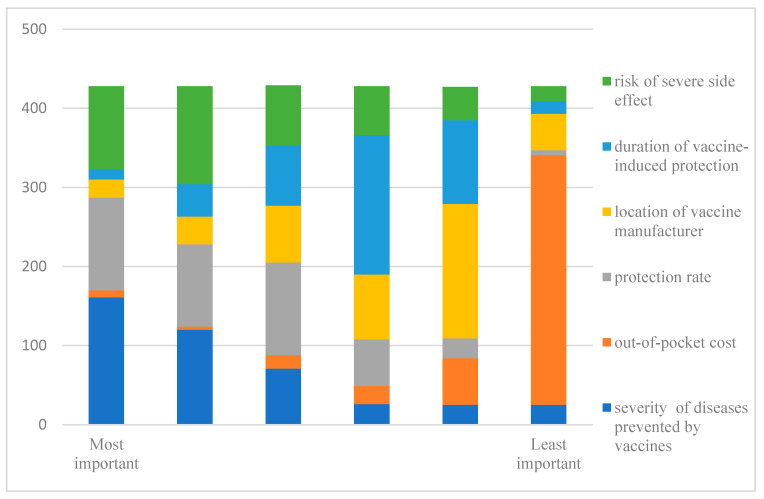
Importance rating of attributes.

**Table 1 vaccines-08-00687-t001:** Attributes and attribute levels for discrete choice experiment (DCE) choice questions.

Attributes	Attribute Levels
The severity of diseases prevented by vaccines (mortality)	1%
	5%
	10%
	15%
Protection rate prevented by vaccines	65%
	80%
	95%
Duration of vaccine-induced protection	1 year
	5 years
	10 years
The risk of severe side effects	Low risk
	Moderate risk
	High risk
Location of vaccine manufacturer	Domestic
	Imported
The out-of-pocket cost of a vaccine	0 Yuan
	150 Yuan
	300 Yuan

**Table 2 vaccines-08-00687-t002:** An example of a discrete choice question (translated version).

Vaccine Attributes	Vaccine A	Vaccine B
Severity of diseases prevented by a vaccine	15%	5%
Protection rate prevented by vaccines	80%	65%
Duration of vaccine-induced protection	1 year	5 year
Risk of severe side effects	Moderate risk	High risk
Location of vaccine manufacturer	Domestic	Imported
Out-of-pocket cost of the vaccine	0 yuan	150 yuan
Which vaccine would you prefer?	☐	☐
In reality, would you vaccinate your child with the option you chose above?	☐ YES ☐ NO	

**Table 3 vaccines-08-00687-t003:** Socio-demographic characteristics of the study population.

Characteristics	All (*N* = 580)		Parents Who Passed the Consistency Test (*N* = 428)		Parents Who Failed the Consistency Test (*N* = 152)	
	Mean	SD	Mean	SD	Mean	SD
Age (years) ^a^	31.22	0.21	31.27	0.24	31.04	0.44
Household size ^a^	4.56	0.05	4.58	0.06	4.53	0.10
Monthly income (RMB) ^a^	12539.2	530.68	12762.24	666.22	11911.18	763.45
Monthly expenditure (RMB) ^a^	7215.10	223.37	7141.47	247.43	7422.41	492.06
Child’ age ^a^	2.03	0.05	2.09	0.06	1.85	0.09
	N	%	N	%	N	%
**Relation to the child** ^b^						
Mother	474	81.72	351	82.01	123	80.92
Father	106	18.28	77	17.99	29	19.08
**Ethics**						
Han	549	94.66	403	94.16	146	96.05
Minority	31	5.34	25	5.84	6	3.95
**Child’s gender** ^b^						
Male	302	52.07 *	236	55.14 *	66	43.42 *
Female	278	47.93 *	192	44.86 *	86	56.58 *
**One child** ^b^						
Yes	261	45.00	193	45.09	68	44.74
No	319	55.00	235	54.91	84	55.26
**Child health** ^c^						
Very good	276	47.59 *	195	45.56 *	81	53.29 *
Good	242	41.72 *	180	42.06 *	62	40.79 *
Fair or poor	62	10.69 *	53	12.38 *	9	5.92 *
**Job** ^b^						
Working	387	66.72	292	68.22	95	62.50
Non-working	193	33.28	136	31.78	57	37.50
**Region** ^b^						
Urban	355	61.21	256	59.81	99	65.13
Rural	225	38.79	172	40.19	53	34.87
**Education levels** ^c^						
Primary or below	8	1.38	7	1.64	1	0.66
Junior or senior	247	42.59	188	43.93	59	38.82
College and above	325	56.03	233	54.43	92	60.52

Note: 1. a—Student’s t-test, b—χ^2^ test, c—Wilcoxon rank-sum test. 2. * *p* < 0.05.

**Table 4 vaccines-08-00687-t004:** Mixed logit model results with only the main effects.

Attributes	β	SE ^†^	SD	SE ^‡^
Non-vaccination	−3.336 ***	0.4464	4.361 ***	0.3548
Protection rate prevented by a vaccine (ref: 65%)				
80%	0.504 ***	0.0757	0.044	0.1407
95%	1.230 ***	0.0882	0.696 ***	0.1125
Risk of severe side effect event (ref: high)				
moderate	0.793 ***	0.0782	0.243	0.1596
low	1.667 ***	0.1131	1.407 ***	0.1156
Location of vaccine manufacturer (ref: domestic)				
Imported	−0.152 *	0.0638	0.740 ***	0.0904
Duration of vaccine-induced protection (ref: 1 year)				
5 years	0.310 ***	0.0732	0.017	0.1317
10 years	0.642 ***	0.0925	0.939 ***	0.1133
Out-of-pocket Cost	−0.001 **	0.0003	0.003 ***	0.0004
Severity of diseases prevented by vaccines (per 1%)	0.003	0.0088	0.136 ***	0.0108
Log likelihood	−3030.5396			
AIC	6101.079			
BIC	6250.286			
Respondents, n	428			
Observations, n	12840			

Note: 1. β—coefficient, SE ^†^—standard error of coefficient, SD—standard deviation, SE ^‡^—standard error of SD, ref—reference, AIC—Akaike information criterion, BIC—Bayesian information criterion. All attributes except for cost and severity of diseases prevented by vaccines were coded as dummy variables. 2. A total of 598 parents enrolled in the survey and 580 completed the majority of the questionnaire. Respondents (428) who passed the consistency test were included in the main effects DCE result reported in this table. 3. * *p* < 0.05; ** *p* < 0.01; *** *p* < 0.001.

**Table 5 vaccines-08-00687-t005:** Results of mixed logit model with main effects and interactions.

Attributes	β	SE	*p*-Value	95% CI	
Non-vaccination	−3.112	0.638	<0.001	−4.361	−1.862
Protection rate prevented by a vaccine (ref: 65%)					
80%	0.501	0.075	<0.001	0.355	0.647
95%	1.386	0.115	<0.001	1.160	1.611
Risk of severe side effect event (ref: high)					
moderate	0.794	0.077	<0.001	0.643	0.944
low	1.609	0.201	<0.001	1.215	2.003
Location of vaccine manufacturer (ref: domestic)					
Imported	−0.403	0.091	<0.001	−0.582	−0.224
Duration of vaccine-induced protection (ref: 1 year)					
5 years	0.315	0.072	<0.001	0.174	0.457
10 years	0.517	0.096	<0.001	0.329	0.706
Out-of-pocket Cost	−0.001	0.000	<0.001	−0.002	−0.001
Severity of diseases prevented by vaccines (per 1%)	0.003	0.011	0.799	−0.018	0.024
**Interaction terms**					
Non-vaccination * working	0.683	0.622	0.272	−0.536	1.902
Non-vaccination * age (>30 years old)	0.584	0.458	0.203	−0.315	1.482
Non-vaccination * father	−1.576	0.585	0.007	−2.723	−0.428
Non-vaccination * rural	−1.283	0.500	0.010	−2.263	−0.304
Non-vaccination * education level (college and above)	0.280	0.501	0.576	−0.701	1.261
Non-vaccination * health state (fair/poor)	−1.899	0.854	0.026	−3.573	−0.224
Lowest risk of severe side effect * education level (college and above)	0.445	0.193	0.021	0.068	0.823
Lowest risk of severe side effect * rural	−0.299	0.190	0.116	−0.671	0.074
Lowest risk of severe side effect * age (>30 years old)	−0.155	0.187	0.406	−0.521	0.211
90% protection rate * age (>30 years old)	−0.362	0.143	0.012	−0.642	−0.081
Protection duration of 10 years * father	0.433	0.204	0.034	0.033	0.833
Imported vaccine * education level (college and above)	0.468	0.124	<0.001	0.225	0.712
Log likelihood	−3234.7672				
AIC	6100.574				
BIC	6339.304				
Respondents, n	428				
Observations, n	12840				

Note: 1. CI—confidence interval. * — the multiplicative relationship which represents the interaction effect of two variables. All attributes except for cost and severity of diseases prevented by vaccines were coded as dummy variables. 2. Interaction terms were treated as fixed effect variables, and the others as random effect variables.

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
