# Peer review of "Parental Vaccine Preferences for Their Children in China: A Discrete Choice Experiment"

_vaccines, 2020, doi:10.3390/vaccines8040687_

Round 1
Reviewer 1 Report
Gong T et al have interviewed about 600 parents from ten different provinces in China and elicited parental vaccine preferences for their children. They report that except for severity of disease all other vaccine attributes were significant. Risk of severer side effect and protection rate of vaccine were found to be the most important factors in parents’ decision towards vaccination. They also report that younger parents had strong preference for vaccine efficiency, while less educated parents preferred domestic vaccines and lower risk for side-effects. The authors employed a discrete choice experiment analyses methodology to realize their study objectives and ran conditional and mixed logit models to obtain study results.
Though study was aptly designed using the DCE approach, there a few major and minor issues that need to be addressed to improve the manuscript.
Major Comments:
- Study background failed to provide a proper context for the need for such a study. Usually studies of this sort are conducted in the context of a new vaccine or in communities with long history of being under-immunized. Neither is the case here.
It is known that childhood immunization in China is almost universal (ref: Chung HJ, Han SH, Kim H, Finkelstein JL. Childhood immunizations in China: disparities in health care access in children born to North Korean refugees. BMC Int Health Hum Rights. 2016;16:13. Published 2016 Apr 13. doi:10.1186/s12914-016-0085-z) and no new vaccine is being added to the current vaccination program. Please comment and revise background/introduction section
- The authors have to keep in mind that the readership is global and hence need to elaborate the study setting further. Most readers will have no clue as to what the authors mean when they are talking about “recent vaccine incidents in China”. Please revise background section.
- Method, Lines, 127-130: I am not sure interviewing parents waiting in lines to get their child vaccinated will make for a representative sample of parents, as one will not find in those lines parents who are vaccine hesitant or resistant to vaccinations. How can the authors claim there are no selection bias issues? The multi-stage sampling should have gone all the way down to residential areas and eligible families should have been interviewed in their households.
Minor Comments:
- Abstract: Please incorporate any revisions made to the other sections in the updated abstract.
- Introduction, Lines 41-59: The authors should have presented current vaccination coverage statistics for the vaccines that they want to see increase in uptake. Even better if this data is China-specific.
- Introduction, Lines 58-59: See Major comment 2. Please revise
- Method, Lines 95-99: Please rephrase these lines with the original formula and elaborate on the calculations. Please mention what each stat represents Ex. what does 500, 4, and 2 represent?
- Method, Lines 113-115: Can you please provide a reference for the steps related to “exclude those who fail consistency test”. It would be interesting to see if results would be any different if the full-set of 580 responses were analyzed
- Method, Line 138: Student’s t-test. Please correct.
- Results, Line 158: Please rephrase “of which 1 and 17 parents were excluded………”
- Tables: Please align the rows properly; SDs were not presented in Tables 4, S1 and S2 yet the foot notes mentions SD; Many more interaction terms had p < 0.05 why weren’t all chosen to be highlighted.
Author Response
Response letter to Reviewer 1 Comment
We would like to thank reviewers for their helpful and constructive suggestions on our study, and the manuscript has been carefully revised. In particular, according to two reviewers’ suggestions, we analyzed full sample data and the results were showed in table S3. We have provided a clean version and a version with tracked changes. Please find a point-by-point response to the reviewer’ comments below.
Major Comment Point 1:Study background failed to provide a proper context for the need for such a study. Usually studies of this sort are conducted in the context of a new vaccine or in communities with long history of being under-immunized. Neither is the case here.It is known that childhood immunization in China is almost universal (ref: Chung HJ, Han SH, Kim H, Finkelstein JL. Childhood immunizations in China: disparities in health care access in children born to North Korean refugees. BMC Int Health Hum Rights. 2016;16:13. Published 2016 Apr 13. doi:10.1186/s12914-016-0085-z) and no new vaccine is being added to the current vaccination program. Please comment and revise background/introduction section
Response 1: We thank the reviewer for the comment. We agree with the reviewer that the DCE technique has a strength of eliciting preferences for a new good/service. Meanwhile, it has also been widely used to understand why an optimal uptake rates have not yet been achieved for existing public health programs, such examples include HPV vaccination and cancer screening services.
In the revised manuscript, we have enriched the 2nd paragraph of the Introduction. In China, some vaccines are mandatory, and children have to vaccinated all these mandatory vaccines before entering primary school. Meanwhile, there are other vaccines which are recommended (e.g. rotavirus vaccine, seasonal influenza vaccine, and pneumococcal conjugate vaccine). Although these vaccines are regarded to be highly beneficial, the uptake rates are often very low (e.g. about 6% and 0.7% for seasonal influenza vaccine and 13-valent pneumococcal conjugate vaccine). Meanwhile, Changchun Changsheng vaccine incident in China came into public insight and caused serious vaccine hesitancy in 2018. Under this context, we conducted this study to elicit the stated preference of parents to vaccinate their children, and therefore we chose general vaccines rather than a specific vaccine.
Point 2: The authors have to keep in mind that the readership is global and hence need to elaborate the study setting further. Most readers will have no clue as to what the authors mean when they are talking about “recent vaccine incidents in China”. Please revise background section.
Response 2: We thank the reviewer for these suggestions. We agree that more detail about vaccine incidents in Changchun City in China in 2018 should be provided. More information and a clear explanation have now been added to the “Introduction” section.
Point 3: Method, Lines, 127-130: I am not sure interviewing parents waiting in lines to get their child vaccinated will make for a representative sample of parents, as one will not find in those lines parents who are vaccine hesitant or resistant to vaccinations. How can the authors claim there are no selection bias issues? The multi-stage sampling should have gone all the way down to residential areas and eligible families should have been interviewed in their households.
Response 3: We appreciate the reviewer’s insight on this representativeness issue. We have revised the corresponding sentences to explain why potential sample selection is not a big worry in this study.
Please line 196-199: Given parents are required to take children to vaccination sites for mandatory vaccines and high vaccination rates have been achieved for these mandatory vaccines, i.e. above 95% for DTaP and hepatitis B vaccines [32], the potential sample selection bias for this recruitment strategy was low.
Minor Comment
Point 1:Abstract: Please incorporate any revisions made to the other sections in the updated abstract.
Response 1: We thank the reviewer for the suggestion. The abstract in the revised manuscript has already been updated.
Point 2: Introduction, Lines 41-59: The authors should have presented current vaccination coverage statistics for the vaccines that they want to see increase in uptake. Even better if this data is China-specific.
Response 2:We thank the reviewer for the suggestions. We agree that more information about the current vaccination coverage should be provided. We have now added it to the “Introduction” section.
Please see line 51-53: The current uptake of recommended vaccines has been estimated to be low in China, that it has been reported to be just about 6% and 0.7% for seasonal influenza vaccine and 13-valent pneumococcal conjugate vaccine, respectively [4,5].
Point 3: Introduction, Lines 58-59: See Major comment 2. Please revise
Response 3: We thank the reviewer for the suggestion. We have now made this clearer (Introduction section).
Point 4:Method, Lines 95-99: Please rephrase these lines with the original formula and elaborate on the calculations. Please mention what each stat represents Ex. what does 500, 4, and 2 represent?
Response 4:We thank the reviewer for these suggestions. We agree with the reviewer that this needs a clear explanation. The statement has now been revised (2.2 section).
Please see line 131-135: The rules of thumb proposed by Johnson and Orme suggested that the sample size can be calculated using the equation: N>500c/(t×a), where c indicates the number of analysis cells , t refers to the number of choice tasks and a is the number of alternatives [27].In the main-effects only design, c is equal to the largest number of levels among different attributes in the DCE. In our study, the corresponding values are c=4, t=10 and a=2; therefore, N can be estimated as (500*4)/(10*2)=100.
Point 5: Method, Lines 113-115: Can you please provide a reference for the steps related to “exclude those who fail consistency test”. It would be interesting to see if results would be any different if the full-set of 580 responses were analyzed
Response 5: We thank the reviewer for the comments. We agree that arbitrary deletion of respondents failing consistency test may be inappropriate. Following the reviewer’s comment, we have re-estimated the models for all observations with results reported in Table S3. It suggests that excluding those who failed the rationality test would have little impact on the results.The statement has been deleted and revised accordingly (3.3 section).
Point 6:Method, Line 138: Student’s t-test. Please correct.
Response 6:We thank the reviewer for the suggestion. The statement has now been corrected. Point 7:Results, Line 158: Please rephrase “of which 1 and 17 parents were excluded………” Response 7:We thank the reviewer for the suggestion. The sentence has now been revised (3.1 section) Point 8: Tables: Please align the rows properly; SDs were not presented in Tables 4, S1 and S2 yet the foot notes mention SD; Many more interaction terms had p < 0.05 why weren’t all chosen to be highlighted.Response 8:We thank the reviewer for the suggestions. Tables have now been revised and we have deleted SDs mentioned in the foot notes of Table 4, S1 and S2. Following reviewer 2’ comment, more interaction terms with p<0.05 have now been highlighted in the abstract section.
Reviewer 2 Report
This is a very interesting study of parental preferences for vaccines in China. The results can help inform vaccine policy. I have a few comments and suggestions:
What are the results if all observations are included? Lancsar and Louviere (2007) suggest “irrational” responses also contain information and suggest not dropping them. Comparing analyses with and without such responses would be interesting.
More context is needed for the results to be useful. Are vaccines mandatory in China? If so, how are mandates enforced? If not, what pressures are placed on parents to vaccinate?
Does the study sample reflect the Chinese population? For example, 61% of the study sample lived in cities; does 61% of the total Chinese population live in cities?
More detail about the Changchun Changsheng vaccine incident would be helpful. Add a sentence or two around line 255 to explain what happened as well as the Shandong illegal vaccine sales.
Along those lines, how is vaccine injury treated in China? Can parents obtain assistance from the government or the vaccine manufacturer if a vaccine injures their child? Or do the costs of vaccine injury fall to the parents?
Lancsar E, Louviere J. 2007. Deleting ‘irrational’ responses from discrete choice experiments: a case of investigating or imposing preferences? Health Economics 18: 797-812.
Author Response
Response letter to Reviewer 2 comments
We would like to thank reviewers for their helpful and constructive suggestions on our study, and the manuscript has been carefully revised. In particular, according to two reviewers’ suggestions, we analyzed full sample data and the results were showed in table S3. We have provided a clean version and a version with tracked changes. Please find a point-by-point response to the reviewer’ comments below.
Point 1:What are the results if all observations are included? Lancsar and Louviere (2007) suggest “irrational” responses also contain information and suggest not dropping them. Comparing analyses with and without such responses would be interesting.
Response 1:We thank the reviewer for the suggestions. Reviewer 1 also asked the similar question. Based on two reviewers’ suggestions, we have re-estimated the models for all observations with results reported in Table S3. It suggests that excluding those who failed the rationality test would have little impact on the results.
Point 2:More context is needed for the results to be useful. Are vaccines mandatory in China? If so, how are mandates enforced? If not, what pressures are placed on parents to vaccinate?
Response 2:We thank the reviewer for these suggestions. We agree that more information about the context should be provided and we have now added relevant information to the “Introduction” section.
Please see line 49-55: In China, several vaccines are mandatory for children and covered by China’s National Immunization Programe (NIP), e.g. diphtheria-tetanus-acellular-pertussis (DTaP) vaccine, measles-mumps-rubella (MMR) vaccine, and hepatitis B vaccine. There are also recommended but not mandatary vaccines, such as rotavirus vaccine, seasonal influenza vaccine, and pneumococcal conjugate vaccine. Mandatory vaccines are free to the public and government-funded, whilst recommended vaccines are normally self-paid by parents.
In China, children have their own child's vaccination certificate. When children are vaccinated all mandatory vaccines, they will get admission to primary school with the child’s vaccination certificate.
Point 3: Does the study sample reflect the Chinese population? For example, 61% of the study sample lived in cities; does 61% of the total Chinese population live in cities?
Response 3: We appreciate the reviewer’s insight on this representativeness issue. First of all, we agree this needs a clear explanation. We have revised the corresponding sentences.
Please see line 196-199: Given parents are required to take children to vaccination sites for mandatory vaccines and high vaccination rates have been achieved for these mandatory vaccines, i.e. above 95% for DTaP and hepatitis B vaccines [32], the potential sample selection bias for this recruitment strategy was low.
However, there is no way for us to figure out the proportion of parents in China based on all characteristics. Following the reviewer’s comment, we decided to choose the data of total Chinese population. This information and reference are now added to the manuscript (3.1 section and reference 35).
Please see line 230: more than half resided in the urban area (61%), which was close to the proportion (60%) of urban population in China [35].
Point 4:More detail about the Changchun Changsheng vaccine incident would be helpful. Add a sentence or two around line 255 to explain what happened as well as the Shandong illegal vaccine sales.
Response 4:We thank the reviewer for these suggestions. Reviewer 1 also asked the similar question. We agree that more detail about the Changchun Changsheng vaccine incident should be added and a clear explanation for Shandong illegal vaccine sales should be provided. The information has now been added to the “Introduction” section.
Please see line 66-70: The Changchun Changsheng vaccine incident involved: i) manufacturing and selling substandard DTaP vaccines and ii) illegal production of freeze dried rabies vaccines [11], while the Shandong illegal vaccine sales referred to questionable vaccines involving 24 provincial-level regions which were produced by licensed manufacturers but not transported or stored properly[12].
Point 5: Along those lines, how is vaccine injury treated in China? Can parents obtain assistance from the government or the vaccine manufacturer if a vaccine injures their child? Or do the costs of vaccine injury fall to the parents?
Response 5: We thank the reviewer for the interesting question. In China, Vaccine Administration Act (reference 51) voted in 2019 stipulated specific measure. A compensation system was implemented for abnormal responses to vaccination, e.g. death, severe disability, organ tissue damage, etc. The bearer of compensation costs depends on whether the vaccine involved is covered by government-funded Expanded Program on Immunization. If a mandatory vaccine causes injuries or other adverse events, governments will take care of the compensation. If it is a recommended vaccine, vaccine manufactures will pay the compensation.
This information is not very relevant to our study. Therefore, we did not provide it.
Round 2
Reviewer 1 Report
Gong T et al have done a good job in responding to the reviewers’ comments. A few additional issues that need to be addressed to improve the manuscript.
Major Comments:
- It appears that the authors have agreed to my argument that excluding certain responses is inappropriate and has no methodological basis to it nor a proper prior reference. In such an instance it is more appropriate that the main findings presented in the paper would be based on results from whole set analyses (n=580) rather than from the subset (n=428).
- I thank the authors for providing more information on mandatory and non-mandatory vaccines in China and low coverage for non-mandatory vaccines. It helped provide a context as to why the study was conducted. Through this study design do the authors think they were able to identify key barriers responsible for very low coverage seen in non-mandatory vaccines? Of the attributes surveyed, only the cost of vaccine appears to be a modifiable factor/attribute while the rest of the attributes of the vaccine or disease are relatively non-modifiable. It would be nice to see some mentioning of how the study findings can be used/leveraged to promote vaccination rates of the non-mandatory vaccines.
- The authors mention in the discussion that majority responded with a “YES” response for the question: In reality, would you vaccinate your child the option you chose above? How do the authors explain this disconnect between high frequency of “YES” and actual poor vaccination rates.
Minor Comments:
- Abstract: Line 24: Please revise as “vaccine-related incidents”
- Abstract: Line 36-37: Please revise based on minor comment no. 5
- Introduction; Line 71-72: Please revise as “arrived at similar”
- Results; Table 3. The layout (the columns) of this table differs from other tables. Was this table a direct reproduction of the results table from the STATA software? It is not clear what the second-set of SD and SE columns represent. Please clarify.
- Results; Table 4
Lowest risk of severe side effect* education level (college and above) 0.445 0.193 0.021 0.068 0.823
Imported vaccine*education level (college and above) 0.468 0.124 <0.001.
Line 223. Please revise “pf” as “of”. Also, I think these two coefficients were interpreted differently. “Less-educated parents preferred domestic vaccines and the lowest risk pf severe side effect.” Both coefficients were had + sign but you flipped “imported” to “domestic” for one but left “lowest risk of severe side effect”.
- Results: It would be recommended to have to more interpretations of coefficients in the text. At least one per table.
- Discussion; Line 249 and Line 250: Please revise these two sentences.
- Discussion; Line 287: Please revise as “lower health literacy”
- Suggest replacing the term “taste differences” throughout the manuscript.
Author Response
We are grateful to the reviewers for careful reading of the paper and helpful suggestions. We have made substantial changes to our manuscript following the reviewers’ comments. A clean version and a version with tracked changes were provided. Please find a point-by-point response to the reviewers’ comments below
Comment: Gong T et al have done a good job in responding to the reviewers’ comments. A few additional issues that need to be addressed to improve the manuscript.
Response:We thank the reviewer for the positive feedback on the revised manuscripts. Please find our clarifications to your comments below. We have also revised the manuscript.Major Comments:
Comment 1: It appears that the authors have agreed to my argument that excluding certain responses is inappropriate and has no methodological basis to it nor a proper prior reference. In such an instance it is more appropriate that the main findings presented in the paper would be based on results from whole set analyses (n=580) rather than from the subset (n=428).
Response 1: We agree with reviewer that we should also report the full sample results in the paper given the preferences from those who failed the rational test may also be valid. Accordingly, we have included the new supplementary table (Table 3) in the revised manuscript and the new results are largely comparable with what have been reported in the main paper. However, we prefer to stick with the subsample who passed the rational test as the key study sample. This decision follows the common procedure in the DCE literature in which the rational test was included. Some examples please see the following published DCE studies:
- Brown DS, Poulos C, Johnson FR, Chamiec-Case L, Messonnier ML. Adolescent girls' preferences for HPV vaccines: a discrete choice experiment. Adv Health Econ Health Serv Res. 2014;24:93-121. PMID: 25244906.
- Ngorsuraches S, Nawanukool K, Petcharamanee K, Poopantrakool U. Parents' preferences and willingness-to-pay for human papilloma virus vaccines in Thailand. J Pharm Policy Pract. 2015 Jul 22;8(1):20. doi: 10.1186/s40545-015-0040-8. PMID: 26199734; PMCID: PMC4509725.
- Wang B, Chen G, Ratcliffe J, Afzali HHA, Giles L, Marshall H. Adolescent values for immunisation programs in Australia: A discrete choice experiment. PLoS One. 2017 Jul 26;12(7):e0181073. doi: 10.1371/journal.pone.0181073. PMID: 28746348; PMCID: PMC5528895.
- Jiang S, Gu Y, Yang F, et al. Tertiary hospitals or community clinics? An enquiry into the factors affecting patients' choice for healthcare facilities in urban China[J]. China Economic Review, 2020:101538.
Comment 2: I thank the authors for providing more information on mandatory and non-mandatory vaccines in China and low coverage for non-mandatory vaccines. It helped provide a context as to why the study was conducted. Through this study design do the authors think they were able to identify key barriers responsible for very low coverage seen in non-mandatory vaccines? Of the attributes surveyed, only the cost of vaccine appears to be a modifiable factor/attribute while the rest of the attributes of the vaccine or disease are relatively non-modifiable. It would be nice to see some mentioning of how the study findings can be used/leveraged to promote vaccination rates of the non-mandatory vaccines.
Response 2: In our study, the process for identifying and selecting the most relevant attributes for parents’ vaccination decision for their children has followed the recommended guidelines, including both literature review and qualitative interviews. In addition to the cost attribute which the reviewer has already mentioned, other attributes are also relevant and played an important role in parental vaccination decision-making. For example, in the background section we explained that owing to the recent vaccine scandals in China, it is unclear whether parents would still choose domestic vaccines and this was specifically investigated as an attribute. Furthermore, our results found that for parents in China, vaccine safety (e.g. side effects) plays a more important role for effectiveness when choosing a vaccine. This information would clearly be relevant for vaccine manufacturers when developing new vaccines (which are normally non-mandatory when just entering the market).
Comment 3: The authors mention in the discussion that majority responded with a “YES” response for the question: In reality, would you vaccinate your child the option you chose above? How do the authors explain this disconnect between high frequency of “YES” and actual poor vaccination rates.
Response 3: In the manuscript we have explained that in China albeit a high vaccination rate was achieved for mandatory vaccines the take up rate was low for the voluntary vaccines. This DCE study was designed to understand what characteristics of vaccines are more important in influencing parental decisions for vaccination their children in general, not in particular referring to mandatory or voluntary vaccines. The significant estimated coefficient of the second-stage question reflects that on average respondents more prefer to vaccinate their children based on the vaccines described in this DCE.
Minor Comments:
Comment 1: Abstract: Line 24: Please revise as “vaccine-related incidents”
Response 1: We thank the reviewer for the helpful suggestion, and this statement have been revised.
Comment 2: Abstract: Line 36-37: Please revise based on minor comment no. 5
Response 2: We thank the reviewer for the suggestion. The sentence has now been corrected.
Comment 3: Introduction; Line 71-72: Please revise as “arrived at similar”
Response 3: We thank the reviewer for the suggestion. We have corrected this phrase.
Comment 4: Results; Table 3. The layout (the columns) of this table differs from other tables. Was this table a direct reproduction of the results table from the STATA software? It is not clear what the second-set of SD and SE columns represent. Please clarify.
Response 4: We thank the reviewer for these suggestions. The mixed logit model differs from the conditional logit that to incorporate the preference heterogeneity, we will estimate a distribution for each attribute levels (rather than only mean coefficients). The standard deviation is used to capture the distribution. We have re-labelled the SE more clearly in the revised tables.
Comment 5: Results; Table 4
Lowest risk of severe side effect* education level (college and above) 0.445 0.193 0.021 0.068 0.823
Imported vaccine*education level (college and above) 0.468 0.124 <0.001.
Line 223. Please revise “pf” as “of”. Also, I think these two coefficients were interpreted differently. “Less-educated parents preferred domestic vaccines and the lowest risk pf severe side effect.” Both coefficients were had + sign but you flipped “imported” to “domestic” for one but left “lowest risk of severe side effect”.
Response 5: We thank the reviewer for these suggestions. We have now corrected this sentence. “Well-educated parents preferred imported vaccines and the lowest risk pf severe side effect.”
Comment 6: Results: It would be recommended to have to more interpretations of coefficients in the text. At least one per table.
Response 6: We thank the reviewer for the suggestion. More interpretations of coefficients have been added to 3.3 section.
Comment 7: Discussion; Line 249 and Line 250: Please revise these two sentences.
Response 7: We thank the reviewer for the suggestion. We have now revised.
Comment 8: Discussion; Line 287: Please revise as “lower health literacy”
Response 8: We thank the reviewer for the suggestion. This statement has been revised.
Comment 9: Suggest replacing the term “taste differences” throughout the manuscript.
Response 9: We thank the reviewer for the suggestion. “taste preference” have been substituted by “preference heterogeneity”.
Reviewer 2 Report
The authors state that Point 5 is not very relevant to the paper, but I disagree. One reason parents might be concerned about vaccine safety is that they would have to pay medical costs if their child sustains a vaccine injury. The authors could explain the vaccine injury compensation program around line 246. As far as I can tell from sources available in English, there is no publicly available database that tracks vaccine injury. This lack of information could explain another reason why parents are concerned with vaccine safety. The authors should include the lack of such a database in their article.
Author Response
We are grateful to the reviewers for careful reading of the paper and helpful suggestions. We have made substantial changes to our manuscript following the reviewers’ comments. A clean version and a version with tracked changes were provided. Please find a point-by-point response to the reviewers’ comments below
Comment: The authors state that Point 5 is not very relevant to the paper, but I disagree. One reason parents might be concerned about vaccine safety is that they would have to pay medical costs if their child sustains a vaccine injury. The authors could explain the vaccine injury compensation program around line 246. As far as I can tell from sources available in English, there is no publicly available database that tracks vaccine injury. This lack of information could explain another reason why parents are concerned with vaccine safety. The authors should include the lack of such a database in their article.
Response: We thank the reviewer for the suggestion. Following the reviewer’s comment, we have now provided this information. Please see line 675-678:
“In China, the first Vaccine Administration Act voted by the Standing Committee of the National People’s Congress in 2019 stated that a compensation system should be implemented for abnormal responses to vaccination, and the bearer of compensation costs depended on whether the vaccine involved was covered by the government-funded Expanded Program on Immunization.”
In China, the Vaccine Administration Act claimed that the quality analysis report of abnormal reaction to vaccines should be submitted to the Medical Products Administrations. However, the database is not publicly available. We agree with the reviewer that the lack of information might be one reason why the vaccination safety was the key characteristic influencing parental vaccination decision-making. According to the reviewer’s suggestion, this information has been added to the discussion. Please see line 679-681:
“Meanwhile, the quality analysis report of abnormal reaction to vaccines should be submitted to the Medical Products Administrations [45]. However, the database is not publicly available, which might be one reason why the safety was the decisive factor.”
Round 3
Reviewer 1 Report
I thank the authors for incorporating the changes suggested. Just one comment:
- Please add a couple of references listed in response to my comment 1 to the paper so that the readers too can see that there is a precedence for this approach.
Author Response
We thank the reviewer for the suggestion, and we have now added the references to the manuscript.
Comment 1:Please add a couple of references listed in response to my comment 1 to the paper so that the readers too can see that there is a precedence for this approach.
Response 1: We thank the reviewer for the suggestion. Two references (reference 36 and 37) have been added in 3.3 section.